# Available In Vitro Models for Human Satellite Cells from Skeletal Muscle

**DOI:** 10.3390/ijms222413221

**Published:** 2021-12-08

**Authors:** Cecilia Romagnoli, Teresa Iantomasi, Maria Luisa Brandi

**Affiliations:** 1Department of Experimental and Clinical Biomedical Sciences “Mario Serio”, University of Florence, Viale Pieraccini 6, 50139 Florence, Italy; cecilia.romagnoli@unifi.it (C.R.); teresa.iantomasi@unifi.it (T.I.); 2F.I.R.M.O. Italian Foundation for the Research on Bone Diseases, Via Reginaldo Giuliani 195/A, 50141 Florence, Italy

**Keywords:** satellite cells, skeletal muscle, myogenesis, cell culture, 2D/3D models

## Abstract

Skeletal muscle accounts for almost 40% of the total adult human body mass. This tissue is essential for structural and mechanical functions such as posture, locomotion, and breathing, and it is endowed with an extraordinary ability to adapt to physiological changes associated with growth and physical exercise, as well as tissue damage. Moreover, skeletal muscle is the most age-sensitive tissue in mammals. Due to aging, but also to several diseases, muscle wasting occurs with a loss of muscle mass and functionality, resulting from disuse atrophy and defective muscle regeneration, associated with dysfunction of satellite cells, which are the cells responsible for maintaining and repairing adult muscle. The most established cell lines commonly used to study muscle homeostasis come from rodents, but there is a need to study skeletal muscle using human models, which, due to ethical implications, consist primarily of in vitro culture, which is the only alternative way to vertebrate model organisms. This review will survey in vitro 2D/3D models of human satellite cells to assess skeletal muscle biology for pre-clinical investigations and future directions.

## 1. Introduction

Skeletal muscle is the largest organ in the human body, representing 35–45% of the total mass. It is essential for breathing, posture, locomotion, and whole-body energy homoeostasis [1]. Skeletal muscle also acts as an endocrine organ able to produce and release, during contraction of myofibers, a myriad of secreted proteins, called myokines, which once in the bloodstream participate in the functional regulation of organs and tissues [2].

The importance of healthy skeletal muscle for the maintenance of the entire body is obvious. Skeletal muscle regenerative ability and function become impaired with aging; genetic diseases, such as Duchenne muscular dystrophy; metabolic diseases, such as metabolic myopathies; and acquired diseases, such as cancer cachexia [3,4,5]. Treatment options for these debilitating myopathies, which severely impact quality of life or shorten lifespan, have limited therapeutic efficacy, and therefore the discovery of novel drugs is necessary. In recent years, research has focused on the study of skeletal muscle biology in order to elucidate the molecular mechanisms that regulate muscle regeneration and plasticity.

Drug discovery and development are traditionally performed in 2D cell culture and small animal models before use in clinical trials, but low efficacy is due in part to animal disease models not truly replicating human diseases, drug response, and toxicity [6,7]. Furthermore, important ethical concerns regarding the use of vertebrate model organisms render important the development of functional and reliable in vitro systems to improve outcomes in human patients by identification of novel therapeutics.

This review will provide a holistic survey on research progress, and evolution of available in vitro models of human satellite cells, to assess skeletal muscle biology for pre-clinical investigations.

## 2. Skeletal Muscle Organization and Repair Process

During embryogenesis, mononucleated myoblasts fuse together to develop multinucleated myofibers that are grouped into bundles and are highly oriented with one another to form a single long cylinder. Within the muscle, each cylinder is surrounded by a layer of connective tissue, known as the perimysium, whereas groups of cylinders are surrounded by the epimysium, another layer of connective tissue, which defines the individual muscle [1]. A single muscle fiber has an approximate dimension between 20 and 100 µm in diameter and up to 20–30 cm of length, and is surrounded by a cell membrane called sarcolemma [8]. The important presence of blood vessels ensures the right amount of nutrients to the whole muscle [9,10].

The terminally differentiated myofibers represent the functional contracting units of skeletal muscle and consist of highly organized myofibrils comprised of repeated sarcomere units that contain myosin and actin which form overlaps to permit muscle contraction in a calcium-dependent manner [11].

Skeletal muscle is a tissue with remarkable capability to adapt to physiological changes and to regenerate in response to growth, physical exercise, and injury, preventing skeletal muscle loss through the formation of new muscle fibers. This incredible plasticity to regenerate is attributed to a small population of mononucleated cells that represent from 2% to 10% of all nuclei of a given fiber in healthy adult mammalian muscle. These cells were identified for the first time by Alexander Mauro in 1961, and called satellite cells (SCs), considering their anatomical location on the surface of muscle fibers, between the basal lamina and the sarcolemma of the myofiber [12,13].

SCs are quiescent and mitotically inactive (G_0_ phase) under resting conditions, but they can rapidly re-enter the cell cycle in response to growth signals or after being injured. Once activated, SCs proliferate to supply myoblasts that differentiate into mature muscle cells, fusing into existing muscle fibers to repair, or fuse together, generating large numbers of new multinucleated myofibers within just few days [14].

Upon muscle injury, a subset of activated SCs undergo terminal skeletal muscle differentiation, fortifying myofibers and contributing to muscle regeneration, while others have the capacity to return to quiescence for replenishing the reserve population of SCs, re-establishing their numbers and quiescent state by homing back to highly specialized niches, thus allowing future regeneration [15].

The multistep muscle formation process is elegantly controlled by a complex gene regulatory network. Quiescent SCs express paired box protein 7 (Pax7), a nuclear transcription factor and well-established SC specific marker, associated with SC quiescence (G_0_ phase) and specification [16,17].

During SC activation, specific gene expression, regulated in a temporally organized manner, ensures terminally differentiated myoblasts. In particular, the myogenic regulatory factors (MRFs), belonging to the basic helix-loop-helix (bHLH) family, are an essential group of muscle-specific proteins, which are exclusively expressed in cells committed to the myogenic lineage. These factors include myogenic factor 5 (Myf5), myogenic differentiation 1 (MyoD1), myogenic regulatory factor 4 (MRF4) and myogenin, responsible for acting at multiple time points in the muscle lineage to cooperatively establish the skeletal muscle phenotype. In general, MyoD1 and Myf5 are expressed in proliferating, undifferentiated cells. In contrast, myogenin expression is induced upon early to late muscle differentiation, while MRF4 is expressed throughout myogenesis (Figure 1) [18,19].

SC proliferation and differentiation are also guided by specific signals secreted by muscle stem cell microenvironment. Muscle injuries and damages cause the release of biologically active molecules into the extracellular space. These molecules can be endogenous to the injured tissue itself or synthesized and secreted by other cell types at the wound site, including neutrophils and macrophages [20].

Hepatocyte growth factor (HGF) is a protein bounded to the ECM of uninjured muscle tissue, and it is released following muscle injury. Its mitogenic and morphogenic activities, especially during the initial phase of muscle repair, are considered to be essential for effective muscle regeneration; in fact, it is known to activate quiescent SCs by binding to its receptor c-Met, which is localized to the SCs membrane [20].

Several members of the fibroblast growth factor (FGF) family are expressed in developing skeletal muscle, and FGF-2 and FGF-6 have been demonstrated to play a role in muscle regeneration through the stimulation of the activation and proliferation of SCs [21].

During muscle regeneration, insulin-like growth factor 1 (IGF-1) increases the proliferation potential of SCs by enhancing the expression of intracellular mediators, such as cyclin-D, and it stimulates terminal differentiation by inducing myogenin gene expression [22].

Tumor necrosis factor-α (TNF-α) is a cytokine quickly produced and released following muscle injury, which activates SCs to enter the cell cycle and enhances cell proliferation [23]. Expression of interleukin-6 (IL-6) after muscle injury is similar to that of TNF-α, and it can be a proliferation signal for SCs to replace the destroyed muscle tissue [24].

Recent data suggest that leukemia inhibitor factor (LIF) is produced by the regenerating muscle itself and seems to play a pleiotropic role during muscle regeneration [25].

Platelet-derived growth factor (PDGF) is released from injured vessels, platelets and macrophages, stimulating angiogenesis; it also causes cell migration, stimulating SCs proliferation [20].

Among the major inhibitors of skeletal muscle regeneration are myostatin, transforming growth factor-α and -β1 (TGF-α and TGF-β1) which are all members of the TGF-β superfamily. This family contains many regulatory factors which, depending on the tissue, affect cellular behavior. In skeletal muscle, TGF-β superfamily members have potent inhibitory effects on both muscle development and postnatal regeneration of skeletal muscle. Myostatin is expressed in SCs and myoblasts and its release results in a down-regulation of Pax-7 and Myf-5 and prevents the expression of MyoD-1. Myostatin and TGF-β1 reduce myoblast recruitment and differentiation [20].

SCs are vital to skeletal muscle homeostasis and regeneration throughout life. The number of SCs per myofiber may differ tremendously between muscles, and myofiber ends can have higher SC concentrations than the rest of the myofiber [26]. There are reports of a decline in the number of SCs in an age-associated environment, characteristic of sarcopenia [4,27]. Moreover, the functional performance of SCs may decline with age, resulting in deficient muscle regeneration [28,29,30]. Muscle wasting associated with muscular dystrophy is also thought to lead to the exhaustion of SCs, due to the continuous demand for reparative myogenic cells [3,31]. Therefore, the importance of increasing knowledge and understanding of the regulation of myogenic stem cells and myogenesis is a very challenging field of research that attracts many scientists, with the potential of providing valuable insight into muscle wasting in aging and disease for the development of new therapeutic drugs.

## 3. Models of Culture for Skeletal Muscle Study

SCs are the only robust source of expandable primary myogenic cells in skeletal muscle, since all myonuclei within myofibers are post-mitotic. Here we show available models of human SCs for use in in vitro studies.

### 3.1. Skeletal Muscle Explants

Isolated portions of skeletal muscle have been cultured since the 1930s, and these models have contributed to the fundamental knowledge of muscle contraction and regeneration on which we base our understanding of human muscle function.

There are two main types of explants: (1) isolated intact muscle fibers (myofibers) used predominately to obtain very pure cultures of satellite cells and to study their dynamics, and (2) isolated intact muscles used to study myofiber contractility and signaling. Like other explant models, their unique contribution is the ability to maintain the complex in vivo cell-cell and cell-matrix interactions [32]. However, these kinds of preparations are very delicate and challenging to prepare; indeed, they require myofibers to remain intact and undamaged from tendon to tendon and be sufficiently thin to enable oxygen and nutrients to reach the explant core. Few studies in humans have been reported; in fact, due to the length of human myofibers (up to 30 cm), this is nearly impossible to achieve by standard biopsy, and therefore these models are almost limited to the animal models [33,34,35,36,37,38,39,40,41].

However, Olsson et al. developed a model of dissected intact muscle fibers obtained from human intercostal muscle biopsies during thoracotomies. Biopsies were collected with intact periosteum at both ends to ensure non-disrupted muscle fibers and preserved tendons at both ends and intact single muscle fibers were dissected [42,43].

### 3.2. In Vitro 2D Models for Skeletal Muscle

Traditional 2D cellular models have the ability to completely control the environment of hSCs and, therefore, they represent important tools to assess the effects of specific drugs or growth factors on skeletal muscle differentiation or the impact of genetic manipulation. Moreover, 2D cell cultures are well-established and widely used, also because they are simple to establish and relatively inexpensive.

Alternative procedures to explant cultures include mincing steps of the whole muscle, followed by enzymatic digestion and repetitive trituration of the muscle for breaking both the connective tissue network and the myofibers, thus allowing the release of SCs by their niches (Table 1).

The digestive enzyme used most for muscle dissociation is collagenase type II, alone [44,45,46,47] or in combination with other enzymes, such as dispase or trypsin, enzyme preparations that preserve cell surface antigens, compared to other mixes of proteases, and which are mandatory for further hSC purification [48,49,50,51,52].

This isolation procedure results in a mixed population of SCs and other mononucleated cells, such as fibroblasts, typically present to some degree in the preparation. This can represent a real problem since the fibroblast population increases dramatically during expansion, as fibroblasts proliferate faster than myoblasts and, hence, this overgrowth significantly reduces SC purity. Moreover, fibroblasts show a more permissive culture condition than SCs, indeed they are able to proliferate in low serum media.

The use of a particular coating of dishes can facilitate the separation of cells. Collagen, laminin, and fibronectin are major proteins of ECM found in tissues that play an important role in regulating cell function in vitro and in vivo; in fact, ECM proteins bind to cell surface receptors and activate signaling pathways that regulate cell morphology, attachment, proliferation, differentiation, and apoptosis [53].

Preferential attachment of myoblasts and fibroblasts has been studied extensively, resulting in the development of pre-plating techniques to purify muscle precursors [54,55,56].

Fibroblast numbers can be minimized by pre-plating on uncoated tissue culture dishes, resulting in separation of cells based on faster adhesion kinetics of fibroblasts compared to muscle progenitor cells [50,57,58].

On the other hand, myoblast precursors can grow at a higher rate compared to fibroblasts on laminin-coated surfaces, which can increase the purity level during culture [48,54]. Several reports describe the use of ECM solution as coating for dishes for human satellite cell cultures [51,59]. Another basement membrane preparation widely used as a coating is Matrigel. It is extracted from the Engelbreth-Holm-Swarm mouse sarcoma, a tumor rich in ECM proteins; its major components are laminin, collagen IV, entactin, and heparin sulfate proteoglycan [60,61]. Matrigel can favor good cell density and complex myotube networks [57,62]. Moreover, Matrigel-coated flasks can enhance proliferation, expression of myogenic markers, and fusion capacity, compared to other coatings [63].

In studies where a pure myogenic population is required, further enrichment of satellite cells is needed, and several techniques have been developed. Myoblast cloning is one of these techniques, allowing the isolation of a pure population of muscle cells derived from single cells [64,65]. Another method is cell sorting using fluorescently or magnetically labelled antibodies (FACS and MACS, respectively) directed against specific surface markers that identify human SCs. Several distinct surface marker panels that contain different positive selecting surface antigens have been used to distinguish human muscle satellite cells from other non-myogenic cell types. These markers include CD56 (NCAM), CD29 (β1-integrin), and CXCR4 (CXC chemokine receptor 4), which are efficiently used for human SC purification [52,66,67,68]. Another panel can further separate SCs from hematopoietic and endothelial cells based on the lack of expression of CD45 and CD31, respectively [52,67]. Moreover, human SCs are negative for CD34 surface expression, in contrast to mouse SCs [52,57,69,70]. Isolated cells are then confirmed to be highly purified by detectable Pax7 immunostaining in nearly all cells, and they can be expanded for experiments in a growth media containing a high serum level to stimulate cell proliferation by high doses of growth factors [59].

Any kind of procedures (digestion and sorting protocols) described to isolate SCs which physically separate them from the whole muscle tissue, perturb their native state, bringing an alteration to gene expression [71]. Once the hSCs are outside their natural niche, cells start losing their quiescent status, Pax7 expression decreases, and they start to activate, elongate, and differentiate into muscle progenitors and myoblasts, typically expressing the transcription factor MyoD1, which can fuse together and form multinucleated myotubes [72,73]. Furthermore, activation of SCs through the FACS process that has been described to affect cell viability and leads to changes in metabolic pathways [74].

As shown, multiple methodologies have been developed to isolate SCs from skeletal muscle tissue. The use of such methods, associated with appropriate culturing conditions, is necessary to isolate SCs from human biopsies. The choice of methods largely depends on the isolation scale (small or large scale) and subsequent experiments.

Sub-confluent cell culture (60–70% before cell passage) is a requisite condition to support cell proliferation and prevent spontaneous cell fusion events during expansion [75,76]. However, with serial in vitro passaging, SCs become senescent and their ability to terminally differentiate decreases, variation in phenotype is amplified and DNA damage accumulates. Moreover, SCs lose the capacity to engraft in skeletal muscle upon transplantation [70,77,78].

The two major mechanisms responsible for the replicative senescence seen in human myoblasts are (i) activation of the p16-mediated cellular stress pathway, and (ii) the progressive erosion of telomeres at each cell division until they reach a critical length that will trigger p53 activation and cell-cycle exit [79,80]. It has been shown that introduction of the telomerase catalytic subunit (hTERT) cDNA in combination with the expression of cyclin-dependent kinase (CDK)-4 is required to successfully overcome cellular senescence in human myoblasts: hTERT elongates the telomere while CDK-4 blocks the p16-dependent stress pathway [81].

With this technique, Mamchaoui et al. were able to produce reliable and stable immortalized cell lines from human myoblasts isolated from biopsies of different muscular dystrophies through the immunomagnetic cell sorting system with anti-CD56 microbeads, resulting in robust in vitro models that can also be implanted in vivo [82]. The immortalized cell lines maintained their myogenic signature and the expression of the myogenic markers desmin, CD56 and MyoD1. Moreover, they were able to differentiate and fuse into myotubes after five days in differentiation conditions by expressing MHC. These cellular models, overcoming the problem of limited proliferation present in myoblasts, provide powerful tools for the scientific community investigating pathological conditions and their mechanisms, and the assessment of new therapeutic strategies.

Human pluripotent stem cells (hPSCs) can represent a promising cell source for regenerative medicine and drug discovery in pathologies of muscle diseases. Differentiation of hPSCs into skeletal muscle cells can be achieved via small molecules-based protocols or ectopic expression of transgenes and allow us to have a virtually unlimited number of cells from a minimally invasive source [14,83,84,85]. Different strategies to obtain functional Pax7+ SCs from hPSCs has been attempted by pairing differentiation protocols with exogenous Pax7 cDNA overexpression [85,86,87].

In recent years, advances in genome-engineering technologies have established the type II clustered regularly interspaced short palindromic repeats (CRISPR)/Cas9 system as a programmable transcriptional regulator capable of targeted activation or repression of endogenous genes [88]. In a study by Kwon et al., they demonstrate that endogenous activation of the Pax7 transcription factor, by this technology, results in stable epigenetic remodeling and differentiates hPSCs into skeletal myogenic progenitor cells which are more proliferative compared to the one’s obtained with exogenous overexpression of Pax7 cDNA, and they can maintain Pax7 expression over multiple passages in serum-free conditions while preserving the capacity for terminal myogenic differentiation [89]. Moreover, transplantation of human myogenic precursors derived from endogenous activation of Pax7 into immunodeficient mice resulted in a great number of human dystrophin+ myofibers compared with exogenous Pax7 overexpression, demonstrating the utility of CRISPR-based activation of gene networks governing progenitor cell specification as a potential strategy for cell therapy and regenerative medicine [89].

### 3.3. In Vitro 3D Models of Skeletal Muscle

Traditional 2D cell culture systems generate short-term models for experiments, while long term ones are often prevented due to the detachment of myotubes [87]. Moreover, 2D models produce developmentally immature myotubes, with limited physiological and translational relevance, and they hinder myotube contraction, a cause for rigid 2D substrates, making it impossible to assess contractile function, a property more representative of in vivo muscle physiology and pathology than variation in proteins and gene expression [87].

To overcome the limitations of 2D cellular models, in vitro 3D skeletal muscle systems have been developed over the last 30 years, in order to recreate the structural organization and functions of adult muscle that are essential for muscle contraction and functionality [9,90]. Tissue engineering is aimed at performing specific mechanical/structural/biochemical functions using cells on a specific support system to create a biomimetic muscle microenvironment [91]. Important requirements of engineering functional skeletal muscle are the recreation of highly aligned myofibrils with myosin/actin filaments, and the formation of contractile myofibers. This approach requires the generation of cell-tissue engineered constructs that can maintain and favor cellular proliferation and differentiation activity and, traditionally, it can be developed considering two different models: (i) scaffold-free/self-assembled skeletal tissue, and (ii) scaffold-based skeletal tissue (Figure 2).

#### 3.3.1. Scaffold-Free Approaches

Scaffold free approaches regard the creation of a 3D system, starting from the seeding of myogenic cells in 2D monolayers under conditions that promote the synthesis and secretion of sufficient ECM proteins, which can self-assemble in 3D tissue after detaching from a 2D surface, to form cylindrical tissue bundles or free-floating planar tissue sheets.

These methods typically require the preparation of a protein-coated polydimethylsiloxane (PDMS) surface of the culture dishes, which permits the pinning of anchors of the 3D structure and the production of the passive tension to induce cellular alignment [92]. The PDMS surface can also be coated with laminin to support cell adhesion [93]. A mixture of myogenic cells and fibroblasts can seed, in order to secrete more ECM proteins and ensure sufficient deposition to enable self-assembly, in general after 35 days [94]. After cells arrive at confluence on the culture surface, the growth medium is switched to a low-serum differentiation medium to start myoblast fusion, resulting in a compact cell layer that gradually detaches from the dish and self-assembles around the anchors in the dish to create a cylindrical 3D structure, termed myooid, that contracts spontaneously at approximately 1 Hz and can be stimulated electrically, producing force [95,96,97,98]. To improve this model and accelerate myoblast fusion and myotube alignment, aligned micropatterned surfaces are used to promote muscle differentiation [93].

Furthermore, detachment of self-organized monolayer muscle cell sheets can be generated using the thermoresponsive polymer, poly(*N*-isopropylacrylamide), as coating [99,100,101,102]. When sufficient ECM is produced, cell sheets can be detached from culture plates by lowering the temperature, and layered with other muscle, vascular, or neuronal cell sheets to generate relatively thick tissue sheets, although the contractile forces of these tissues have not been reported [99,102].

Another recent approach of the 3D system without a scaffold is the creation of a self-organized skeletal muscle organoid which consists of a 3D structure generated from freshly isolated myogenic precursors [103,104,105]. Following isolation from skeletal muscle tissue, and initial seeding of mononucleated cells suspension on plate, after three to five days, some small clusters of a few cells form free-floating rounded sphere-like structures which propagate in vitro. They can be passaged every 20–30 days by dissociating the spheres larger than 100 µm with collagenase or dispase and then re-plating them at a density from 1 to 10 × 10^5^ cells/mL. The replated cells can form new free-floating spheres over a period of several days. Myosphere culture characterization has evidenced that they contain myogenic cells expressing Pax7, Myf5, and MyoD1, and cells derived from myospheres behave similar to primary myoblasts, as well as forming multinucleated myotubes when cultured adherently, even if some differences in terms of proliferation rate, differentiation capacity and phenotype are present [106]. The main advantage of the 3D formation of myospheres is the fact that they can maintain a pre-myogenic state in culture over time, starting from little manipulation of the freshly isolated SCs, exhibiting cell-cell interaction and spatial organization between cell types closer to that observed in tissues.

Despite the benefits that scaffold-free approaches can bring for multiple assays, such as in functional studies and drug screening, and the importance of the production of its own secreted ECM, bypassing the variability of commercially available ECM proteins, the longer time to tissue formation (around 35 days), small tissue size, and challenges with scale-up have limited the use of these methods compared to other scaffold-based approaches.

#### 3.3.2. Scaffold-Based Approaches

Scaffold-based approaches utilize specific constructs constituted by biomaterials which must be biocompatible and contain high surface area to allow cellular adhesion and colonization; moreover, they have to promote and facilitate nutrient diffusion and should be resorbable once they have served their purpose of providing a primary structure for the developing tissue [107].

The most common 3D systems used for skeletal muscle models are hydrogels derived from natural ECM proteins, such as collagen, laminin, and fibrin [91,108]. In fact, naturally derived hydrogels (collagen, fibrin) support high density and 3D spread of muscle cells, unidirectional alignment through the application of geometric constraints, and macroscope tissue contractions [107,109,110]. Alternatively, synthetic polymers, such as poly-L-lactic acid, polylactic-glycolic acid, and polyurethane or decellularized muscle, can be used as scaffolds for muscle tissue generation [91,111,112]. The main advantage of the synthetic scaffolds compared to the naturally derived materials is the fact that they can be precisely characterized and fabricated with excellent control over physical and chemical properties [91,112,113].

During 3D tissue fabrication, initially developed by Rhim et al. [114], myogenic precursor cells and gel mixture is cast into silicone tubing or PDMS mold coating with a pluronic solution to prevent cell adhesion. Cylindrical molds are, in general, used to generate a simple, cable-like tissue geometry with porous felts at the ends of the mold that serve to anchor the hydrogel once it is compacted, and provide passive tension that enables cellular forces to remodel the hydrogel and maintain the tissues under tension to promote cellular alignment, rapid fusion, and muscle hypertrophy. Once the mixture is molded and the hydrogel is polymerized, the construct is placed in the growth medium for a few days. It is then switched to the differentiation medium to induce fusion of myoblasts into myofibers, followed by progressive structural and functional maturation [9,115].

The first 3D tissue-engineered muscles use collagen type I hydrogels, since this is the most abundant ECM protein in skeletal muscle [109,116,117,118,119]. This approach was initially used to generate tissue from animal cells, but it has also been demonstrated that embedding primary human myoblasts within collagen hydrogel permits the formation of the engineered muscle tissue [118,120,121].

Collagen type I hydrogel has several limitations due to its stiffness and ease of rupture. It is not easily remodeled and does not stimulate ECM secretion, which is understandable if we consider in vivo fibrosis and muscle dysfunction, in which there is an excess of collagen type I and poor regeneration and function of native muscle [97,122,123].

Moreover, collagen hydrogel generates low contractile force compared to fibrin, having adverse effects on muscle functional maturation. This is in part attributable to the lack of expression of α7 and αv integrin binding between mature myotubes in this kind of hydrogel [124]. In fact, in native muscle, myofibers interact with basal lamina, which is rich in laminin and collagen type IV but not collagen type I [125].

To ameliorate collagen type I hydrogel, and improve muscle structure, Matrigel is added, even though it generates lower contractile forces compared to fibrin [126].

Unlike collagen type I, which was initially used by most laboratories, and considering all of the limitations concerning its use, fibrin is a preferred choice as a matrix for tissue engineering of skeletal muscle.

Fibrin is the major component of blood clots and plays a fundamental role in wound healing, where it is replaced over time by secreted ECM [127]. Despite the disadvantages of fibrin in terms of lot-to-lot variability and its variable degradation rate, which can be overcome by lot testing and control fibrinolysis using cross-linkers or anti-fibrinolytic compounds, thanks to its properties of being extensively remodeled and degraded, and hence replaced by endogenously derived ECM, it is considered an ideal substrate for tissue engineering [128]. Fibrin also promotes angiogenesis and the extension of neuronal axons, which are fundamental for the formation of a fully functional engineered muscle for in vivo applications [129].

Human myoblasts differentiate in a fibrin hydrogel generated tissue with muscle-like stiffness and an elastic modulus of 12 kPa, similar to that of native muscle, which facilitate myogenic gene expression and promote myogenic differentiation [130,131,132]. Moreover, to improve engineered muscle structure of the tissue, and contractile force, Matrigel can be supplemented to the hydrogel [115,124,133].

Once the engineered skeletal muscle tissue is formed, it is cultured in specific conditions to improve myofiber maturation and favor their survival. Traditionally, biophysical and biochemical stimulation is applied in order to mimic the natural environment and stimulate cell proliferation and differentiation. In particular, to promote cell alignment and enhance fusion of myoblasts, improving maturation and myofiber hypertrophy, mechanical or electrical stimulations can be applied to the 3D constructs to mimic growth or exercise, or neuronal input to muscle [116,118,134,135]. The addition of growth factors, such as IGF-1, TGF-β or Wnt, can be useful to further promote the maturation and contractile function of myofibers [136].

Despite the incredible results of the engineered muscle tissues, recapitulating many of the morphological and functional features of native skeletal muscle, they generate contractile properties that are inferior to those of native adult muscle, revealing an incomplete muscle maturation [133,137].

The new tissue must be biocompatible, innervated and vascularized. Since native skeletal muscle tissue and functionality is derived by the cooperation of different cellular types, increasing lineage complexity by using multicellular types typical of native muscle (vascular network, tendon, neuronal input) other than purely myogenic precursors in the 3D skeletal muscle constructs represents a strategy to design a more physiological model of skeletal muscle [138,139]. In a recent study, the creation of an artificial muscle was described, obtained by differentiating myogenic progenitors in fibrin hydrogels and, subsequently, embedding the muscle fibers in hydrogel containing endothelial cells and myofibroblasts [140]. Other studies create multilineage artificial muscle anchored at two attachment points using muscle progenitor cells co-cultured with motor neuron spheroids on top of muscle bundles [132,141].

Vascularization, in particular, represents an important characteristic for tissue-engineered skeletal muscle for their long-term survival and integration in vivo. Native skeletal muscle is highly vascularized to provide oxygen and nutrient supply required to support the high metabolic demands induced during growth and by muscle contraction. Hypoxia and impaired cell survival typically occur at a diffusion distance of 150–200 µm from blood vessels or in culture media and limit tissue-engineered muscle size; in fact, beyond this distance, the necrotic core is formed at the tissue center [142]. The formation of stable vasculature required the inclusion of supporting cell types such as fibroblasts, pericytes and smooth muscle cells [143]. However, the ability of the formed vascular network to increase nutrient and oxygen delivery and enable survival of larger muscle constructs in vitro has yet to be shown [115].

Currently, the precise spatial patterning of cells the and extracellular matrix to create compartmentalized 3D constructs is currently best achieved by the 3D bioprinting technique. This is an additive manufacturing technology that fabricated tissue analogues by stacking living cells and biomaterials layer by layer and helps the cell growth and signals through modulation of the cell-cell interaction and cell-matrix interaction. This method represents a powerful tool for tissue engineering because it can easily fabricate bulk and complex tissue, mimicking the structure and endowing cells with a biomimetic 3D microenvironment in the desired shape and with the appropriate architecture of native muscle tissue [144,145]. Various 3D bioprinting methods, including inkjet, microextrusion and laser-assisted methods, are highly used because they ensure high cell viability. A recent work by Kim et al. shows that bioprinting human skeletal muscle constructs are able to form multi-layered boundles with aligned myofibers and that neural cell integration into the construct accelerates functional muscle regeneration, increasing long-term survival and neuromuscular junction formation in artificial skeletal muscle in vitro [139]. Another work by Choi et al. showed that using 3D bioprinting with human primary myogenic progenitors in strips encapsulated by a human endothelial cell lines in parallel with microchannels enhances the vascularization and functional recovery upon implantation in a rodent model [146].

## 4. Future Directions

Significant progress in skeletal muscle models has been made over the last two decades. However, this field of research remains a challenge for researchers, and we expect that in the near future new methodological advancements will take place.

3D engineering muscle tissue, compared to traditional 2D monolayers, provides longer-term culture, improved myofiber maturation, and the ability to measure functional properties of the skeletal muscle. Therefore, engineered muscle is a promising goal for in vitro studies of human skeletal muscle regeneration and function as well as disease. Methods for fabrication of scaffolds that can easily fabricate bulk and complex tissue, mimicking the structure and endowing cells with a favorable microenvironment for proliferating and differentiating which can bring about the realization of engineered skeletal muscles with more appropriate architecture of native muscle tissue, are urgently needed. Moreover, strategies to enhance cellular complexity of the constructs, which include the incorporation of tendons, moto-neurons, myofibroblasts, and other supporting cell types that are required to generate more biomimetic engineered muscles, will enable the creation of more physiological models to develop new in vitro platforms of skeletal muscle tissue for predictive drug screening and disease modeling.

The development of improved methods for the maintenance and expanse of hSCs in vitro, without losing self-renewal or myogenic capacity, represents the principal issue to be addressed to obtain sufficient numbers of regenerative myogenic cells. Alternative sources of myogenic progenitors should be extensively considered and investigated, since the possibility of using human immortalized myogenic precursor cells or hPSCs would allow us to obtain a virtually unlimited number of cells from a minimally invasive source [132,147]. Moreover, the opportunity to use them as xenografts, engrafted into a rodent host, represents a valid preclinical tool to better investigate muscle repair and regeneration and mechanisms of muscular diseases in order to more accurately predict potential novel cell therapies [139,148,149]. Several advances in xenografting human-derived muscle cells have been applied to study and treat different muscular dystrophies [148,150,151] and they set the foundation for designing future experiments and to encourage investigators to test new approaches that may be applicable for skeletal muscle regeneration.

## Figures and Tables

**Figure 1 ijms-22-13221-f001:**
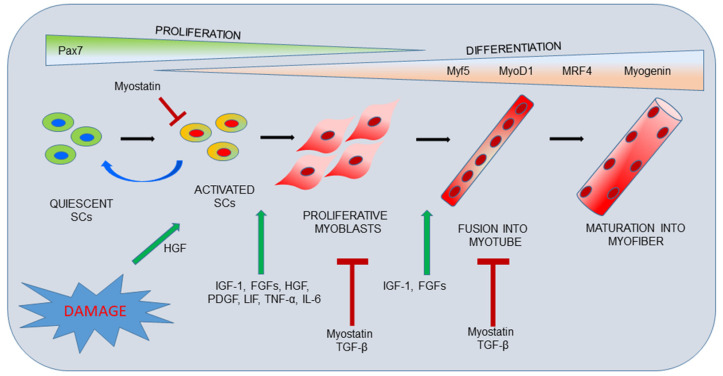
Skeletal muscle regeneration. Activation and differentiation processes of SCs are finely controlled by a genetic cascade involving Pax7 and the myogenic regulatory factors (Myf5, MyoD1, MRF4 and Myogenin), which drive every step of skeletal muscle regeneration, up to the formation of new muscle fibers. Activated SCs can retain Pax7 expression and return to a quiescent state to contribute to the replenishment of the SC pool for future muscle regeneration. Specific growth factors modulate SC activity and they are released from a number of tissues after tissue damage, and they are responsible to modulate SC proliferation and differentiation.

**Figure 2 ijms-22-13221-f002:**
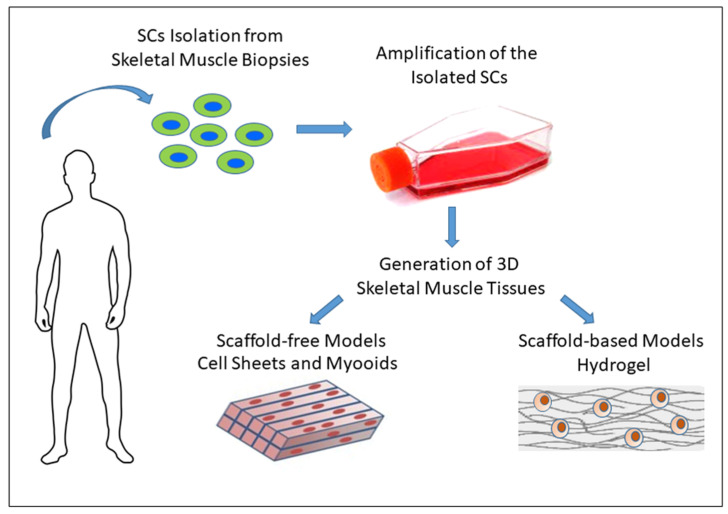
3D cell-tissue engineered approach for in vitro skeletal muscle investigations. Engineered skeletal muscle models are based on 3D systems of cells scaffold-free or generated by combining cells and an appropriate scaffold.

**Table 1 ijms-22-13221-t001:** Different methodologies to establish, isolate, and characterize hSCs from skeletal muscle biopsies.

Digestive Enzyme	Isolation Method	Plate Coating	Markers	Reference
Collagenase II	Enzymatic digestion	-----	Desmin+	[33]
Collagenase II	FACS	Collagen I	CD56+	[34]
Collagenase II	Enzymatic digestion	Collagen I	Pax7+ MyoD1+	[35]
Collagenase II	Enzymatic digestion	-----	-----	[36]
Collagenase II/ Dispase	Pre-plating to remove fibroblasts	Laminin	MyoD1+	[37]
Collagenase II/ Dispase	MACS	-----	CD56+	[38]
Collagenase II/ Trypsin/EDTA	Pre-plating to remove fibroblasts	-----	CD56+ CD29+ CD34-	[39]
Collagenase II/ Dispase	FACS	ECM proteins	CD34- CD45- CD31-	[40]
Collagenase II/Trypsin	MACS/FACS	Laminin	CD56+ CD29+ CXCR4+	[41]
Trypsin	Serial Plating	Laminin/ Collagen I	Desmin+	[43]
Trypsin/EDTA	Cell Cloning	ECM proteins	-----	[45]
Collagenase II	Enzymatic digestion	Matrigel	Pax7+	[46]
Trypsin/EDTA	Enzymatic digestion	ECM protein	-----	[48]
Trypsin/EDTA	Cell Cloning	Collagen I	-----	[53]
Trypsin/EDTA	FACS Cell Cloning	Collagen I	CD56+	[54]
Trypsin/EDTA	FACS	Collagen I	CD56+	[57]
Trypsin/EDTA	Pre-plating to remove fibroblasts	ECM proteins	-----	[73]

## Data Availability

Not applicable.

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
