# Peer review of "Available In Vitro Models for Human Satellite Cells from Skeletal Muscle"

_ijms, 2021, doi:10.3390/ijms222413221_

Round 1

Reviewer 1 Report

In this review the authors have made a comprehensive survey of “in vitro 2D/3D models of human satellite cells to assess skeletal muscle biology for pre-clinical investigations and future directions”. This will complement several reviews which have appeared recently on this hot topic. Platforms using engineered human muscle tissue are being increasingly used in drug discovery and therefore this review is timely. However there is some room for improvement.

The section “Skeletal muscle organization and repair process” can be shortened and more focused. In its present form it is not clear what this adds to the subject matter which is “available in vitro models for human satellite cells from skeletal muscle”.

In addition, ln 68/69 I am surprised that single muscle fibers are only 1cm in length, are the authors sure that they could not be longer in humans for example in the vastus lateralis? Ln 91 I think should be plasticity. Ln 93 If they concentrate on humans then in healthy adults the value will be between 2 and 5% rather that 10% which is rather high?

Ln158 – the authors should be aware that explant cultures are a means of obtaining very pure cultures of human satellite cells. It is an alternative method of obtaining cells from healthy or pathological tissue used by many researchers in particular for small muscle biopsies where enzymatic digestion would give poor yields of cells.

Single fiber cultures – this technique has been mainly described for rodents where the fibers can be isolated intact when muscles can be dissected tendon to tendon. This section should be rewritten if the focus as the title states is ‘in vitro models for human satellite cells”. In human there have been a few studies. It is difficult to obtain muscle biopsies for this technique. One situation would be to use intercostal muscles where the fibers can be dissected intact. As stated they are difficult to maintain for any length of time in culture making them as stated by the authors “practically unusable for high throughput screening, making large-scale experiments in human tissue almost infeasible” therefore is this section really necessary?

In vitro 2D models – satellite cells can be isolated either from explant cultures or by enzymatic digestion. This should be added.

In general, human myoblasts can be easily purified using beads coated with CD56. This is current practice now even for cell therapy. FACS is generally avoided for human cells since this will affect cell viability. Precoating and preferential pre-plating was used in the past but is no longer used since beads coated with specific antibodies are available. Different methods of coating are more important to maintain the differentiated myotubes attached to the dish. In general, this section is rather misleading and should be rewritten This was in fact the precursor to using different methods of embedding muscle cells to bioengineer 3D cell cultures.

A section could be added here on immortalized human cell lines which have been used for both drug screening as well as developing gene therapy approaches including CRISP Cas 9 (see article from Gersbach’s lab).

Limitations of 2D Cellular Models.

In this section beware because digestion and sorting are not only a limitation of 2D but also 3D because these digested and sorted cells will be seeded in the different matrixes to make the 3D cultures to. This should be clarified or removed. In addition, it is evident that if the cells are isolated and amplified their gene expression profile will obviously be different from the non-dividing quiescent cells. Therefore, I would suggest removing this paragraph or put it in another context.

Ln 265 not quite clear what sort of regenerative experiments are referred to hear. It maybe useful to add a phrase about the telomere driven limit of proliferation in human cells which need to be monitored so that work will not be done on senescent or presenescent cells. This will be overcome when doing preclinical studies using immortalize human cell lines.

Ln 269-271 – this is only the case when the experiments are carried out using senescent cells in which case many parameters may change – remove or modify

The big problem with 2D cultures is their lack of maturity and organization. When cells are embedded in a 3D scaffold they are able to align, to form a correct basal lamina to organize the system of T tubules and contract allowing measurement of muscle function.

In Vitro 3D Models of Skeletal Muscle

It is not clear to me how maintaining the niche like environment and maintaining the cells in a more primitive state is pertinent with the formation of the 3D cultures where cells are in general seeded at high density in an ECM matrix? In general, this introductory paragraph could be shorter and more focused.

Future Directions

Maybe a section could be added on Xenografting human muscle to look at human muscle regeneration and pathology (for example see articles from K Wagner T Partridge and R Bloch).

Beware it is stated “This review will survey in vitro 2D/3D models of human satellite cells to assess skeletal muscle biology for pre-clinical investigations and future directions” but in many cases the experiments described are in rodents this should be specified in the text.

By shortening and focusing the text this will leave room to add a little more information about the application for disease modeling and drug testing.

Reviewer 2 Report

The review-manuscript by Romagnoli and Brandi nicely describes the current literature on vitro models for human satellite cells of native skeletal muscle. The manuscript is well written, reads nicely, and includes most literature on the particular scientific research area. This reviewer has only minor suggestions for improvements.

  • Lines 98-103. Please expand and include a more detailed description on the signals that regulate proliferation and differentiation in SCs. In addition, please include this information in figure 1.
  • Recently, a study evaluated and described viability of isolated and incubated skeletal muscle from mice (PMID: 34057444). A discussion of these findings should be placed in section 3.1 of the manuscript.
  • Lines: 181-189: Please include a short discussion on the Pro’s and Con’s for the different isolation procedures mentioned in table 1.
  • If 3D models for skeletal muscle become as big as native skeletal muscle in the future, it may be speculated that these models will also suffer from lack of perfusion / supply of nutrition (decreased viability) as isolated small native muscles (e.g., Soleus and EDL). Please include a short discussion of this issue.

Reviewer 3 Report

The authors, Romagnoli and Brandi, wrote a comprehensible and exhaustive review entitled “Available in vitro models for human satellite cells from skeletal muscle”, in which known in vitro models for the study of human satellite cells (SCs) have been described.

The review reports an overview of skeletal muscle features and their importance for body function, focusing on skeletal muscle organization, repair process, SCs and myogenic regulatory factors (MRFs). The review also include the description of different methods for studying skeletal muscle biology in vitro, covering the most relevant approaches used in myology research.

Overall, the review is well-written and well-structured and summarizes current knowledge on the topic. However, before publication, the review would gain substantially from addressing specific points listed below, line by line.

  • 40 - I suggest to remove “tissue” from this sentence.
  • 42 - I suggest to remove “can” from this sentence.
  • 42, 43 - I suggest to indicate Duchenne Muscular Dystrophy as a genetic disease and add specific metabolic myopathies as examples of muscle metabolic disease.
  • 55, 56, 59 - Latin terms should not be written with italicized text.
  • 68, 69 - Replace “has an approximate dimension of 100 μm” with “ranges from 20 to 100 μm”. This information has been reported in “Quantitative human physiology: an introduction” (2017), Joseph Feher, 2nd
  • 91 - Replace “pasticity” with “plasticity”.
  • 104 - Replace “Some daughter SCs continue to differentiate” with “Upon muscle injury, a subset of activated SCs undergo terminal skeletal muscle differentiation”.
  • 111-114 - Replace the sentence from “recognized” to “inhibiting differentiation” with “and well-established SC specific marker, associated with SC quiescence (G0 phase) and specification”.
  • 129-133 - The functional characterization of SCs in response to tissue repair and aging is well addressed also in Tierney MT et al., Cell Stem Cell 2018.
  • 139 - Minor point: a dot is missing after “Figure 1”.
  • 141 - Uniform the nomenclature of genes/proteins.
  • 149 - Latin terms should not be written with italicized text.
  • 184 - Replace “to allow” with “thus allowing”.
  • 191 - Replace “fibroblastic cells” with “other mononucleated cells, such as fibroblasts”.
  • 194 - Please, add also this notion: “Moreover, fibroblasts show a more permissive culture condition than SCs, indeed they are able to proliferate in low serum media”.
  • 223,224 - Please, add “(FACS)” and “(MACS)” acronyms where appropriate.
  • 278, 282, 314, 416, 441, 487, 490 - Latin terms should not be written with italicized text.
  • 342 - Replace “polimer” with “polymer”.
  • 347 - Replace “have not be” with “have not been”.
  • 389 - Replace “glicolic” with “glycolic”. Add also “acid” where appropriate.

Further recommended consideration: please uniform the nomenclature of human genes and proteins (e.g. Figure 1 caption). I suggest to briefly discuss about 3D bioprinting in the section concerning in vitro 3D models, citing the main papers about it.

Please specify (Paragraph 2, lines 115-123) the expression level of MRFs in terminally differentiated myotubes/myofibers.

Please, specify the schedule of passages for culture methods described in the paragraphs 3.1, 3.2, and comment on feasibility in terms of time.

Concerning references, follow MDPI citations style guide.

Round 2

Reviewer 1 Report

The authors have done a nice job in revising their manuscript and have taken all of my comments into consideration in this revised version.

Reviewer 3 Report

The revised version of the review, written by Romagnoli and colleagues, have been exhaustively implemented and corrected. However, I have listed (line by line) few words and sentences that need to be corrected before publication.

  • 130 – “Myf5” has been misspelled to “Mrf5”.
  • 134 – 135 I suggest to reformulate this sentence as follows: “SC proliferation and differentiation are also guided by specific signals secreted by muscle stem cell microenvironment.”
  • 137 – Replace “may” with “can”.
  • 138 – Remove “or may be”.
  • 340 – Replace “described affects” with “described to affect”.
  • 368 – Remove “the”.
  • 409 – Replace “compare” with “compared”.
  • 370 – Replace “inket” with “inkjet”.
  • 757 – Replace “to more accurate predict” with “in order to more accurately predict”.